# SARS-CoV-2 Infection Is an Independent Risk Factor for Decompensation in Cirrhosis Patients

**DOI:** 10.3390/diseases12030046

**Published:** 2024-02-25

**Authors:** Mark Ayoub, Julton Tomanguillo, Carol Faris, Nadeem Anwar, Harleen Chela, Ebubekir Daglilar

**Affiliations:** 1Department of Internal Medicine, Charleston Area Medical Center, West Virginia University, Charleston, WV 25304, USA; julton.tomanguillochumbe@camc.org; 2Department of General Surgery, Marshall University, Huntington, WV 25755, USA; 3Division of Gastroenterology and Hepatology, Charleston Area Medical Center, West Virginia University, Charleston, WV 25304, USA; nadeem.anwar@camc.org (N.A.); harleen.chela@camc.org (H.C.)

**Keywords:** COVID-19, SARS-CoV-2, cirrhosis, decompensation, GI bleed, varices, ascites, mortality

## Abstract

Background: SARS-CoV-2 causes varied gastrointestinal symptoms. Cirrhosis patients face higher mortality rates from it, especially those with decompensated cirrhosis. This study examines SARS-CoV-2’s impact on decompensation in previously compensated cirrhotic patients. Methods: We analyzed the Global Collaborative Network, comprising 98 healthcare organizations across sixteen countries, using TriNetX’s deidentified research database. Compensated cirrhosis patients were split into two groups: one with SARS-CoV-2-positive patients and another testing negative. Using a 1:1 propensity score matching model based on baseline characteristics and comorbidities, we created comparable cohorts. We then assessed decompensation, mortality, and GI bleed at 1 and 3 months. Results: Out of 252,631 identified compensated cirrhosis patients, 27.3% (69,057) tested SARS-CoV-2-positive, while 72.6% (183,574) remained negative. Post PSM, 61,963 patients were in each group. SARS-CoV-2-positive patients showed significantly higher decompensation rates (4.4% vs. 1.9% at 1 month; 6% vs. 2.6% overall). Rates of complications, like ascites, SBP, HE, and HRS, increased notably. Mortality (2.5% vs. 1.7% at 1 month; 3.6% vs. 2.7% at 3 months) and GI bleed (1.3% vs. 0.9% at 1 month; 1.9% vs. 1.2% at 3 months) were also elevated in SARS-CoV-2 patients. Conclusions: SARS-CoV-2 increases decompensation over 2-fold in compensated cirrhosis patients and raises mortality and increases rates of complications at 1 and 3 months.

## 1. Introduction

SARS-CoV-2 infection is associated with a variety of gastrointestinal manifestations. Patients with cirrhosis are particularly interesting, as there is emerging data that cirrhotic patients have an increased mortality rate with SARS-CoV-2 [1]. Mortality rates have also been noted to increase in patients with decompensated cirrhosis (when compared to compensated disease) and increase with the rising Model for End-Stage Liver Disease (MELD) score or Child–Pugh classification. Irrespective of the pre-existing liver disease status, SARS-CoV-2 infection with elevated liver enzymes, especially AST, on presentation, results in significantly higher rates of mortality, intubation rates, and prolonged hospitalization [2]. Patients with cirrhosis are effectively immunocompromised, which makes them prone to infections. This study aims to assess if SARS-CoV-2 infection is associated with new clinical decompensations of the liver with new onset jaundice, ascites, spontaneous bacterial peritonitis (SBP), hepatic encephalopathy (HE), hepato-renal syndrome (HRS), and esophageal variceal bleeding (EVB).

## 2. Materials and Methods

### 2.1. Statistical Analysis

The study was approved by the Institution Board Review Committee at the Charleston Area Medical Center. Written informed consent from patients was waived due to the de-identified nature of the TriNetX clinical database. The TriNetX (Cambridge, MA, USA) database is a global federal research network that combines real-time data with electronic medical records. Our study was conducted using the TriNetX database through the Global Collaborative Network, which comprises 98 Healthcare Organizations (HCOs) from sixteen countries. Adult patients aged ≥18 years with compensated cirrhosis were identified between November 2013 and November 2023. Compensated cirrhosis was identified as patients who had a documented diagnosis of cirrhosis, regardless of etiology, who did not have any prior decompensation. Patients with compensated cirrhosis were identified using the codes from the International Classification of Diseases (ICD)-10. A list of all ICD-10 codes and definitions for the study is highlighted in the Appendix A. Patients with compensated cirrhosis were divided into two groups: patients who tested positive for SARS-CoV-2 infection and patients who tested negative for SARS-CoV-2 infection. This was followed by the propensity score matching (PSM) of both groups to ensure successful and effective balancing. PSM was performed using baseline patients’ characteristics and comorbidities as highlighted in Table 1.

After conducting propensity score matching, an examination of the outcomes was undertaken. Differences in all-cause mortality rates between groups were explored using Kaplan–Meier curves and log-rank tests. For each outcome, risk ratios (RRs) along with their corresponding 95% confidence intervals (CIs) were computed. Statistical significance was established at a *p*-value < 0.05. The statistical analyses were carried out utilizing the TriNetX platform.


### 2.2. Inclusion and Exclusion Criteria

Patients with compensated liver cirrhosis, regardless of etiology, were identified and divided into two cohorts: patients who tested positive for SARS-CoV-2 and patients who tested negative for SARS-CoV-2. SARS-CoV-2 positivity was defined as the presence of IgG or IgM antibodies or SARS-CoV-2 RNA in serum. We excluded any patient who had any decompensation event prior to testing. Decompensation events were identified as the presence of esophageal varices with or without bleeding, hepatic encephalopathy (HE), spontaneous bacterial peritonitis (SBP), jaundice, ascites, or hepatorenal syndrome (HRS). The following outcomes were compared at 1- and 3-month intervals: mortality and decompensation. We also compared each separate decompensation event between the cohorts: jaundice, ascites, SBP, HE, HRS, and esophageal variceal bleeding (EVB).

## 3. Results

### 3.1. Baseline Characterestics

A total of 252,631 patients with compensated cirrhosis were identified and met our inclusion criteria. Patients with compensated cirrhosis who tested positive for SARS-CoV-2 (27.3%%, *n* = 69,057) and patients with compensated cirrhosis who tested negative for SARS-CoV-2 (72.6%, *n* = 183,574) were assessed. Two well-matched cohorts of patients who tested positive for SARS-CoV-2 and those who tested negative (*n* = 61,963/*n* = 61,963) were compared following propensity score matching using baseline patients’ demographics and comorbidities.

Analyses of cohorts’ baseline demographics and comorbidities did not show any significant difference after PSM. The mean age in the group that tested positive was 54 years with a standard deviation of 11.4. Slightly more than half the cohort was comprised of males 56.1%. In the group that tested positive for SARS-CoV-2, chronic obstructive pulmonary disease (COPD) was found in 11.3%, coronary artery disease (CAD) in 11.4%, hypertension (HTN) in 51%, and chronic kidney disease (CKD) in 10.4%. A full comparison of cohorts’ baseline demographics and comorbidities before and after PSM is highlighted in Table 1.

### 3.2. Outcomes

After PSM, we compared different outcomes between the two cohorts. Patients with compensated cirrhosis who tested positive for SARS-CoV-2 infection had a statistically significant higher rate of decompensation when compared to those who tested negative (4.4% vs. 1.9%, *p* < 0.0001 at 1 month and 6% vs. 2.6%, *p* < 0.0001 at 3 months with an overall odds ratio of 2.4). Patients who tested positive for SARS-CoV-2 infection also had a higher mortality rate (2.5% vs. 1.7%, *p* < 0.0001 at 1 month and 3.6% vs. 2.7%, *p* < 0.0001 at 3 months). GI bleed was higher in those who tested positive for SARS-CoV-2 (1.3% vs. 0.9%, *p* < 0.0001 at 1 month and 1.9% vs. 1.2%, *p* < 0.0001 at 3 months).

Subanalysis of decompensation events showed that the rates of ascites, SBP, HE, and HRS were significantly higher in those who tested positive for SARS-CoV-2 infection compared to those who tested negative. The rate of jaundice was not statistically different between the two groups. A summary of the results is shown in Table 2 and graphs showing the outcomes over time are shown in Figure 1, Figure 2 and Figure 3.

## 4. Discussion

### 4.1. Cirrhosis and Immune Dysfunction

Our immune system comprises an innate immune system and an adaptive immune system. Both systems are impaired in patients with cirrhosis [3]. Cirrhosis affects innate immunity by impairing the synthesis and function of various proteins that have bactericidal properties [4]. Monocyte function is one of the cell lines responsible for chemotaxis, phagocytosis, and the production of lysosomal enzymes [5]. Immune paralysis, which is defined as a decreased expression of monocytes, is a known phenomenon in patients with liver disease and it is believed to be the mechanism through which this cell line is affected [6]. Neutrophils, which are the first line against bacterial infections, are also affected in patients with liver disease [3] Neutrophil impairment results in the inability of neutrophils to reach the infection site and a decreased phagocytic ability of the neutrophils toward the infection [3] The liver is the site of clearance of many cytokines. Neutrophil dysfunction leads to a persistent activation of neutrophils, which, in turn, leads to a persistent elevation of cytokines [7] This elevation plays a part in the occurrence of cytokine storm and systemic inflammatory response, which is one of the mechanisms through which SARS-CoV-2 causes liver damage [8].

Adaptive immunity dysfunction is also common on patients with cirrhosis [3]. Patients with cirrhosis have a significant derangement of monocytes and T cells [9]. This is secondary to the prolonged activation of T-cell lymphocytes, which hinders their ability to proliferate after a new insult leading to immunosuppression secondary to the exhaustion of an adaptive immune system [3]. Another affected component in patients with cirrhosis is IgA; however, the mechanism is not fully understood [10].

The complement system plays part in both innate and adaptive immunity and is also affected in patients with cirrhosis [3]. The liver is the synthetic site of complement factors; therefore, low levels of C3 and C4 are seen in patients with cirrhosis [11]. This leads to the increased susceptibility of patients with cirrhosis to bacterial infections. Another mechanism of immunosuppression in patients with cirrhosis is bacterial translocation [12]. Bacterial translocation is the migration of bacteria or their products from intestinal lumen to mesenteric lymph nodes [13]. Multiple risk factors increase the risk of bacterial translocation, which includes impaired humoral and cellular immunity, increased intestinal permeability, and bacterial overgrowth [14]. Bacterial translocation is strongly associated with spontaneous bacterial peritonitis and is another mechanism of how immune dysfunction can lead to hepatic decompensation [12]. A summary of cirrhosis effect on the immune system is highlighted in Figure 4.

### 4.2. Mechanism of SARS-CoV-2-Induced Liver Injury

Patients with SARS-CoV-2 infections have been known to develop a wide variety of gastrointestinal complaints and symptoms, and studies have been performed to describe the extent to which SARS-CoV-2 causes disease [15]. SARS-CoV-2 infection has been shown to cause elevated bilirubin levels, creatinine levels, and decreased albumin, with the concern regarding SARS-CoV-2’s first wave concerning direct hepatic injuries associated with the disease; however, the evidence for a viral cause for hepatic injuries is still missing [15,16,17]. Additionally, elevated aminotransferase levels at the time of admission also correlate with higher mortality [2,18]. Patients with underlying chronic liver disease were of particular interest, as patients with cirrhosis were noted to have worse outcomes with concomitant SARS-CoV-2 infections [1,2,16,19,20,21].

SARS-CoV-2 can cause abnormal liver function tests directly and indirectly in people with a baseline impaired function [22]. The mechanism of hepatotoxicity can be related to the nature of the virus, its consequences, or its treatment. In Wuhan, China, some studies reported an elevation of liver function tests in up to 53% of cases [23]. The pathophysiology behind the hepatotoxic effect of the SARS-CoV-2 virus is believed to be due to the attachment of the S-spike protein to the angiotensin-converting enzyme 2 (ACE2) receptor leading to virus entry into hepatocytes and decreased hepatic function [24,25]. Once bound, it breaks vasoconstrictor angiotensin II into vasodilator angiotensin I, which leads to a drop in blood pressure. These receptors are found in cholangiocytes, hepatocytes, pancreas, and other major organs [26,27]. SARS-CoV-2 also enters endothelial cells and destroys the vascular endothelium. This, in turn, causes vasoconstriction and, furthermore, a procoagulant state [28,29]. The procoagulant state was observed in patients with respiratory failure and it was occurred due to hypercoagulability rather than consumption, as reported by Spiezia [30]. He found that those patients had high fibrinogen and D-dimer levels. He also observed a hypercoagulable platelet profile, which was confirmed by Rampotas and Pavord when they observed platelet aggregation and increased platelet activity when they examined the blood films of patients with SARS-CoV-2 who were mechanically ventilated [31] This was also further confirmed by a systematic review examining the risk of portal vein thrombosis (PVT) in patients with SARS-CoV-2, which found a higher association of PVT in hospitalized patients with SARS-CoV-2 infection [32]. PVT is well known to cause hepatic decompensation in patients with cirrhosis [33].

Another mechanism of SARS-CoV-2-induced liver injury is the occurrence of cytokine storms [8]. SARS-CoV-2 is associated with a strong inflammatory response, which leads to the activation of the pro-inflammatory cascade involving cytokines, such as interleukins 1B, 6, and 18 [34]. Their activation leads to the activation of other interleukins and tumor necrosis factor-alpha. This eventually leads to an elevation of ferritin and c-reactive protein levels and subsequent endothelial damage [35]. Additionally, SARS-CoV-2 is known to cause hypoxia [36]. This leads to a decrease in hepatic perfusion and subsequent hepatocellular hypoxia. This primary hepatic injury leads to elevations in AST and ALT levels [37]. Secondary hepatic injury occurs as a result of an acute inflammatory response in the settings of sepsis and multi-organ failure [37].

### 4.3. SARS-CoV-2 Drug-Induced Liver Injury

Furthermore, many of the drugs used for SARS-CoV-2 treatment have a hepatotoxic effect, which further contributes to its role in decompensation [38]. A systemic review and meta-analysis revealed a pooled incidence of drug-induced liver injury in patients with SARS-CoV-2 of 25.4% [39].

Dexamethasone, which is used frequently for respiratory failure secondary to SARS-CoV-2 infection, is known to cause elevated liver enzymes [40]. Its use was associated with both liver enzyme elevation and the reactivation of chronic Hepatitis B [41] However, their use in those patients improved their mortality rate and respiratory status [42].

Remdesivir inhibits RNA polymerase is metabolized in part by cytochrome P450 in the liver [43]. Multiple studies reported the hepatotoxic effect of Remdesivir, which included significant elevations of ALT and AST leading to the discontinuation of therapy [44,45,46].

Ritonavir is another drug used for SARS-CoV-2 infection. It is a part of the commercially available drug Paxlovid, which is recommended by the Center for Diseases Control and Prevention (CDC) for outpatient use against SARS-CoV-2 [47]. It is extensively metabolized by the cytochrome P450 system [48]. A randomized, controlled, open-label trial including patients with SARS-CoV-2 infections who were severely ill and required hospitalization showed that Ritonavir caused elevated ALT, AST, and total bilirubin levels [49]. Both cholestatic and hepatocellular patterns of liver injury were reported with the use of Ritonavir [48].

Imatinib, which is also used for SARS-CoV-2 infection, is known to cause three forms of liver injury: transient elevation of liver enzymes, acute hepatitis, and reactivation of chronic hepatitis B [50]. There have been multiple case reports of acute liver injury with Imatinib use, some of which resulted in death or liver transplants [51,52].

Baricitinib is used in combination with Remdesevir for SARS-CoV-2 infection. It has been associated with the elevation of liver enzymes, but the data are still limited on the long-term effects [53]. In many large clinical trials on patients with rheumatoid arthritis, Baricitinib was found to cause an elevation in liver enzymes in up to 17% of the patients, which occasionally led to the discontinuation of treatment [54].

Azithromycin is also commonly used for SARS-CoV-2 treatment due to its effect of reducing the severity of lower respiratory tract illnesses [55]. This is thought to be via binding to the ACE2 receptor-SARS-CoV-2 Spike protein complex, which inhibits the initial stages of SARS-CoV-2 replication [56]. Of note, the S-spike protein of SARS-CoV-2 independently causes a strong immune reaction activating both the innate immune system as well as the adaptive humoral system, which further causes liver injury [57]. The mechanism of liver injury is unknown, but the rapid onset suggests a hypersensitivity component [58]. Azithromycin can cause liver injury in the short term and long term [59]. Short-term injury usually occurs within 2–3 days and it is highlighted by a hepatocellular pattern of injury [60]. This hepatocellular pattern can be severe and can lead to acute liver failure. Long-term injury occurs within 1–3 weeks and is highlighted by a cholestatic pattern [61]. This can be accompanied by a long duration of jaundice and the further impairment of liver function.

Acetaminophen, a commonly used antipyretic medication, is also frequently used for SARS-CoV-2 infection. The data for acetaminophen’s hepatotoxic effect are widely available, and its use in patients with SARS-CoV-2 infections can cause mild hepatotoxicity with elevations in ALT and AST levels [62]. The mechanism of hepatic injury is extensively studied. The most well-known form is a serious hepatocellular injury after an overdose; this is generally secondary to the direct effect of such high doses [62].

### 4.4. SARS-CoV-2 and Underlying Liver Disease

Underlying liver disease was observed in up to 11–17% of patients with SARS-CoV-2 infections [2,63]. Patients with an established diagnosis of cirrhosis had increased mortality with acute SARS-CoV-2 infection compared to patients with a mortality of 32% for patients with cirrhosis compared to 8% for those without cirrhosis [17,21]. The risk of mortality in cirrhotic patients with SARS-CoV-2 infections has been shown to increase with the increase in the baseline MELD or Child–Turcotte–Pugh score [15,21,64]. A study in 2020 compared the mortality of patients with SARS-CoV-2 infections; compared to patients without a chronic liver disease, mortality increased by 20.0% in Child–Pugh class-B cirrhosis patients and 38.1% in patients with the Child–Pugh class-C disease [21]. The increase in mortality of patients with SARS-CoV-2 was consistent with an increase in the mortality of patients with decompensated cirrhosis (defined by the presence of ascites, hepatic encephalopathy, spontaneous bacterial peritonitis, and esophageal varices) compared to patients with compensated cirrhosis (absence of factors qualifying for a decompensated status) [64]. These findings are also in line with our study outcomes. Our study showed a 1.5-fold increase in mortality in patients with compensated cirrhosis if tested positive for SARS-CoV-2 infection.

### 4.5. Decompensation of Cirrhosis

Ascites is the most common complication in up to 60% of patients with cirrhosis [65]. In patients with a normal renal function, diuretics can be introduced [65]. However, a large volume of ascites needs to undergo paracentesis to rule out SBP and to provide symptomatic relief. IV albumin should be administered at the time of paracentesis to reduce the risk of hepatorenal syndrome (HRS).

Acute kidney injury (AKI) is common in up to 20% of patients with decompensated cirrhosis [66]. It can occur as a pre-renal AKI, a primary renal disease, or HRS. AKI in those patients is usually multifactorial [66]. The initial treatment for AKI is the correction of volume status with IV fluids or albumin. Diuretics and any nephrotoxic medication should be discontinued.

Gastrointestinal bleed is another cause of decompensation, and variceal bleed accounts for 50% of gastrointestinal bleeds in patients with cirrhosis [67]. Ensuring a patent airway, reliable IV access, and resuscitation are key elements in the initial management of gastrointestinal bleeds in patients with cirrhosis. Fixing concurrent coagulopathy and blood transfusion is recommended [68]. Endoscopy should be performed to achieve bleeding source control [69].

Another cause of decompensation is hepatic encephalopathy, which can be precipitated by infections, constipation, electrolyte abnormalities, sedatives, or gastrointestinal bleeding [70]. Treating hepatic encephalopathy includes treating the underlying cause as well as laxatives. Lactulose can be administered orally or rectally and should be titrated to a goal of about three bowel movements daily [70]. Rifaximin can be added if the response to lactulose is inadequate [70].

### 4.6. Prognosis of SARS-CoV-2 in Patients with Cirrhosis

A large cohort study in England showed that patients with prior liver disease on presentation and admitted with SARS-CoV-2 infection had a higher mortality rate [71]. Another study in China evaluated the characteristics of patients with SARS-CoV-2 infections; of 62 patients, 11% had an underlying liver disease, which was higher than any other comorbidity [72]. Furthermore, patients with non-alcoholic fatty liver disease (NAFLD) had a much higher incidence of liver injury [73,74]. This sheds light on how susceptible this patient population is.

The development of decompensated cirrhosis is a concern regarding SARS-CoV-2 infections due to the increased mortality in this population. In a study from 2020, hepatic decompensation developed in 36.9% of patients with SARS-CoV-2 infections and was strongly associated with an increased risk of death [64]. In an additional study from 2020, decompensation occurred in 46% of patients with cirrhosis [21]. Our study showed a 2.5-fold increase in the decompensation rate in patients with previously compensated cirrhosis if tested positive for SARS-CoV-2 infection, with rates of 4.4% and 6% at 1 and 3 months compared to 1.9% and 2.6%, respectively. Studies since then recommend routine testing for SARS-CoV-2 infection in patients with acute hepatic decompensation, even without respiratory symptoms [16,18,20]. Further information is needed to confirm this trend, with the need for more extensive studies and ongoing study populations outside the index SARS-CoV-2 infection as new variants continue to emerge. Although the SARS-CoV-2 infection may be associated with acute hepatic decompensation, little is known about increased GI bleeding or the development of esophageal variceal bleeding. One review noted GI blood loss in patients with SARS-CoV-2 to be less frequent than other GI manifestations, with 4% of critically ill patients with SARS-CoV-2 documented to have GI blood loss. In addition, esophageal variceal bleeding (EBV) is less common than other events marking hepatic decompensation [64]. In a nationwide study, Gandhi et al. also showed that the presence of SARS-CoV-2 infection in patients with bleeding varices led to a significant delay in endoscopic interventions compared to patients testing negative, which further resulted in increased all-cause mortality and ICU admissions [75].

### 4.7. Strengths and Weaknesses

One of our study strengths was the large sample of patients, which increased the power of the study and allowed generalizability. Another major strength was the selective nature of our inclusion and exclusion criteria. We exclusively included patients with compensated cirrhosis and excluded any patients with a prior history of any decompensation event prior to the study index.

Our study had a few limitations. The nature of our dataset did not allow us to characterize the severity of liver disease, such as the MELD score; however, we attempted to mitigate such a limitation by our selective inclusion and exclusion criteria and by using PSM, which ensured a very similar patient population in the two groups. Another study limitation related to the nature of our dataset was the inability to specify the SARS-CoV-2 strain included in our study.

### 4.8. Summary

The SARS-CoV-2 pandemic has created many challenges and changed the healthcare landscape across the globe. The SARS-CoV-2 infection is known to have caused a wide variety of gastrointestinal symptoms, and studies are ongoing to determine the extent and severity. Patients with cirrhosis have an impaired immune system, which makes them more susceptible to various infections. SARS-CoV-2 causes direct and indirect hepatic injuries. Also, many medications that are used to treat SARS-CoV-2 infection have hepatotoxic effects. Therefore, patients with cirrhosis who end up contracting the SARS-CoV-2 infection have a higher risk of hepatic decompensation. Our study suggests that SARS-CoV-2 infection is an independent risk factor for the decompensation of cirrhosis in previously compensated patients.

## 5. Conclusions

The presence of SARS-CoV-2 infection, regardless of the symptoms, is associated with more than a two-fold-higher rate of decompensation among patients with previously compensated liver cirrhosis. It is also associated with a higher mortality rate at the 1- and 3-month marks from the date of infection. This should encourage healthcare providers to be vigilant to diagnose SARS-CoV-2 as an acute decompensating event, even with a lack of respiratory symptoms.

## Figures and Tables

**Figure 1 diseases-12-00046-f001:**
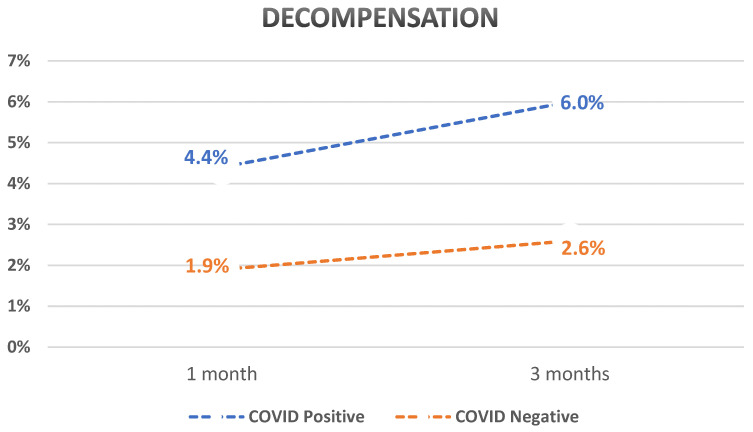
Graph showing a comparison between decompensation rates over time for previously compensated cirrhosis patients who tested positive for SARS-CoV-2 compared to those who tested negative.

**Figure 2 diseases-12-00046-f002:**
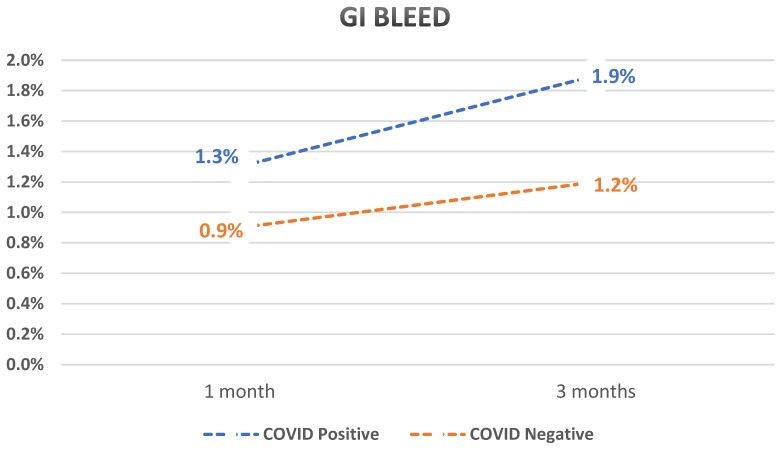
Graph showing a comparison between GI bleed rates over time of previously compensated cirrhosis patients who tested positive for SARS-CoV-2 compared to those who tested negative.

**Figure 3 diseases-12-00046-f003:**
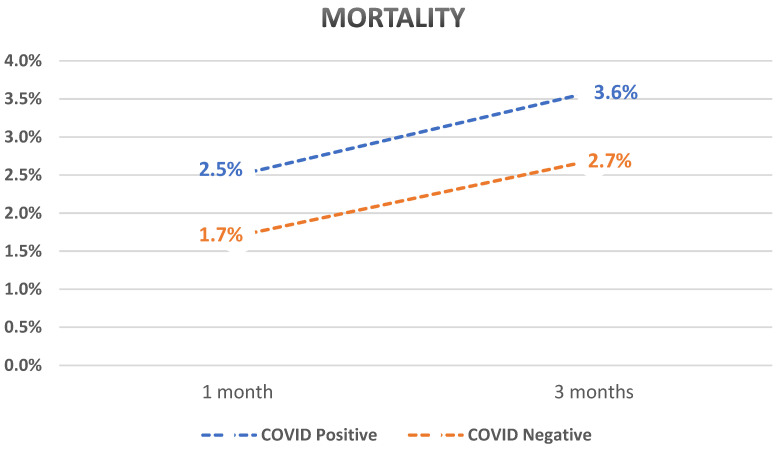
Graph showing a comparison between mortality rates over time of previously compensated cirrhosis patients who tested positive for SARS-CoV-2 compared to those who tested negative.

**Figure 4 diseases-12-00046-f004:**
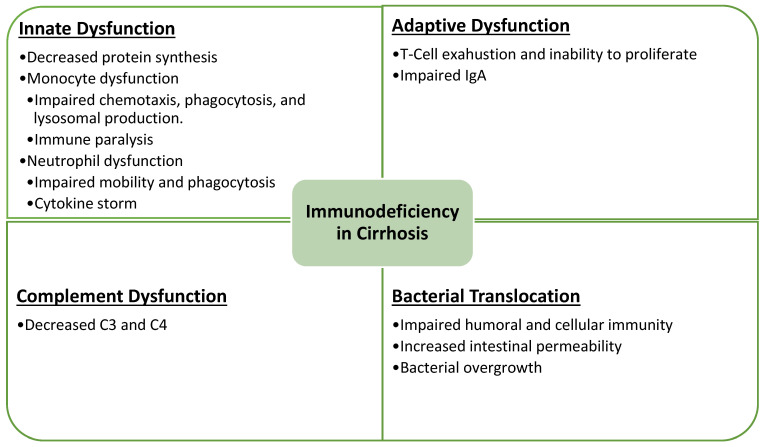
Mechanism of immune dysfunction in patients with cirrhosis.

**Table 1 diseases-12-00046-t001:** Patient characteristics of cohorts.

SARS-CoV-2-Positive and SARS-CoV-2-Negative counts before and after propensity score matching						
Cohort	Patient count before PSM matching			Patient count after PSM matching		
SARS-CoV-2-Positive	8367			5092		
SARS-CoV-2-Negative	8150			5092		
Propensity score density function—before and after matching (SARS-CoV-2-positive—purple, SARS-CoV-2-negative—green)						
	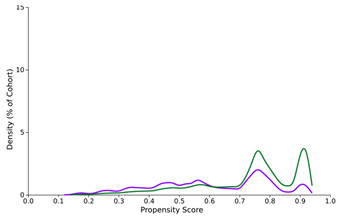			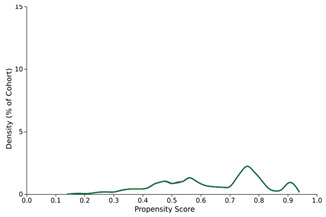		
	Before PSM			After PSM		
	SARS-CoV-2-Positive (*n* = 69,057)	SARS-CoV-2-Negative (*n* = 183,574)	*p*-value	SARS-CoV-2-Positive (*n* = 61,963)	SARS-CoV-2-Negative (*n* = 61,963)	*p*-value
Demographics						
Age at Index Mean ± SD	54.3 ± 11.2	51.4 ± 11.1	<0.001	53.6 ± 11.4	53.2 ± 11	0.08
Female	43.9%	40.6%	<0.001	43.3%	43.7%	0.14
Not Hispanic or Latino	73.5%	51.5%	<0.001	71.7%	70.3%	0.07
Comorbidities						
COPD	14.5%	4.9%	<0.001	10.4%	10.6%	0.33
CAD	15.7%	4.8%	<0.001	11.1%	11.2%	0.54
CKD	14.3%	4.7%	<0.001	10.2%	10.3%	0.41
Hypertension	55.7%	24.5%	<0.001	50.7%	51.8%	0.09
Diabetes Mellitus	33.8%	15.1%	<0.001	29.2%	30.6%	0.1

SD: Standard Deviation; COPD: Chronic Obstructive Lung Disease; CAD: Coronary Artery Disease; CKD: Chronic Kidney Disease.

**Table 2 diseases-12-00046-t002:** Summary of results.

	Outcomes at 1 Month				Outcomes at 3 Months			
	SARS-CoV-2-Positive (61,963)	SARS-CoV-2-Negative (61,963)	OR	*p*-Value	SARS-CoV-2-Positive (61,963)	SARS-CoV-2-Negative (61,963)	OR	*p*-Value
Decompensation	4.4%	1.9%	2.4	<0.0001	6%	2.6%	2.4	<0.0001
Jaundice	0.7%	0.6%	1.1	0.08	0.9%	0.8%	1.1	0.03
Ascites	2.6%	0.4%	6	<0.0001	3.5%	0.7%	5.4	<0.0001
SBP	0.1%	0.02%	4.7	<0.0001	0.17%	0.03%	5	<0.0001
HE	1.9%	0.9%	2.2	<0.0001	2.5%	1.2%	2.2	<0.0001
HRS	0.2%	0.05%	4.3	<0.0001	0.27%	0.07%	4	<0.0001
GI bleed	1.3%	0.9%	1.5	<0.0001	1.9%	1.2%	1.6	<0.0001
Mortality	2.5%	1.7%	1.5	<0.0001	3.6%	2.7%	1.3	<0.0001

OR: Odds Ratio; SBP: Spontaneous Bacterial Peritonitis; HE: Hepatic Encephalopathy; HRS: Hepatorenal Syndrome; GI: Gastrointestinal.

## Data Availability

Available data is presented. Additional data is only available as permitted by third party.

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
