# Peer review of "SARS-CoV-2 Infection Is an Independent Risk Factor for Decompensation in Cirrhosis Patients"

_diseases, 2024, doi:10.3390/diseases12030046_

Round 1

Reviewer 1 Report

Comments and Suggestions for Authors

the authors ăpresenta very inetersting topic, the one regarding the influence of SARS COV2 infection on the outcomes of liver cirrhosis patients. 

Comments on the Quality of English Language

Minor English corrections are needed

Author Response

Thank you very much for taking the time to review this manuscript! I appreciate your time and input.

In response to the Quality of English Language: I agree with your remark. We proofread the article and ran it through grammar and spelling editing services to minimize errors.

Reviewer 2 Report

Comments and Suggestions for Authors

The authors insisted that presence of SARS-COV-2 infection regardless of symptoms is associated with more than two-folds higher rate of decompensation among patients with previously compensated liver cirrhosis. It is also associated with a higher mortality rate at the 1- and 3-month marks from date of infection date.

I have some questions

1. There is no information regarding when this study was conducted or the main infecting strains. It is necessary to consider whether this is a problem with the Delta strain, which is more severely affected by coronavirus infection, or whether the same thing occurs with recently weakened strains.

2. Regarding drug-induced liver damage discussed in the discussion, what is the usage status of antiviral drugs in the patients in this study?

3. It seems that there is no significant difference between the two groups in the patient character cohorts, but is the same true for liver function? Is it possible to add data immediately after infection to Figures 1 and 2?

4. If a patient with liver cirrhosis becomes infected with coronavirus, please discuss clinical countermeasures, such as whether antiviral drugs should be actively used or steroids should be used to suppress the cytokine storm.

Author Response

Dear Reviewer,

Thank you so much for your critical feedback. It was very helpful!

 Below is a point-by-point response to your feedback:

“The authors insisted that presence of SARS-COV-2 infection regardless of symptoms is associated with more than two-folds higher rate of decompensation among patients with previously compensated liver cirrhosis. It is also associated with a higher mortality rate at the 1- and 3-month marks from date of infection date. I have some questions”

  1. There is no information regarding when this study was conducted or the main infecting strains. It is necessary to consider whether this is a problem with the Delta strain, which is more severely affected by coronavirus infection, or whether the same thing occurs with recently weakened strains.

Excellent remark about the Delta strain, unfortunately in this dataset, we were not able to identify each strain as a study variable, we added this into our limitations. 

As for the study timeframe, it was outlined in the methods section. 

  1. Regarding drug-induced liver damage discussed in the discussion, what is the usage status of antiviral drugs in the patients in this study?

We mentioned the usage of antivirals in the discussion section. We have not addressed the usage of antiviral drugs in this particular study, this would be an excellent study topic alone for further studies!

  1. It seems that there is no significant difference between the two groups in the patient character cohorts, but is the same true for liver function? Is it possible to add data immediately after infection to Figures 1 and 2?

We used propensity-score matching to minimize any difference between the groups to ensure very similar cohorts. In terms of liver function, we excluded all decompensation-related events including ascites, etc in both groups trying to minimize true liver function-related difference. 

  1. If a patient with liver cirrhosis becomes infected with coronavirus, please discuss clinical countermeasures, such as whether antiviral drugs should be actively used or steroids should be used to suppress the cytokine storm.

These suggestions were incorporated into the discussion section.

Thank you again for your feedback!

Reviewer 3 Report

Comments and Suggestions for Authors

The manuscript of Ayoub is well-written and the study is important for both the medical specialists and researchers. As a minor comment, I would suggest wide the discussion part and including some animal studies, related to the topic and the studies about how Sars-CoV-2 Spike protein by itself (as an experimental model or vaccination product) can provoke or exacerbate the decompensation. 

Author Response

Thank you very much for taking the time to review this manuscript. I appreciate your time and input!

In response to your comment "I would suggest wide the discussion part and including some animal studies, related to the topic and the studies about how Sars-CoV-2 Spike protein by itself (as an experimental model or vaccination product) can provoke or exacerbate the decompensation"

The S-spike protein plays an important role in the virus-hepatocyte integration. Your comment is very relevant to the study especially for the drug-induced injury portion of the discussion. Therefore, I went ahead and added it under the paragraph 4.3. “SARS-COV-2 Drug-induced Liver Injury” and I also briefly mentioned the mechanism in paragraph 4.2. "Mechanism of SARS-COV-2-induced liver injury". I also included the animal study citation you mentioned.

Again, thank you for your feedback and time!